# Forrest Classification for Bleeding Peptic Ulcer: A New Look at the Old Endoscopic Classification

**DOI:** 10.3390/diagnostics12051066

**Published:** 2022-04-24

**Authors:** Hsu-Heng Yen, Ping-Yu Wu, Tung-Lung Wu, Siou-Ping Huang, Yang-Yuan Chen, Mei-Fen Chen, Wen-Chen Lin, Cheng-Lun Tsai, Kang-Ping Lin

**Affiliations:** 1Department of Internal Medicine, Division of Gastroenterology, Changhua Christian Hospital, Changhua 500209, Taiwan; 91646@cch.org.tw (H.-H.Y.); 180278@cch.org.tw (T.-L.W.); 182972@cch.org.tw (S.-P.H.); 27716@cch.org.tw (Y.-Y.C.); 2General Education Center, Chienkuo Technology University, Changhua 500020, Taiwan; 3Department of Electrical Engineering, Chung Yuan Christian University, Taoyuan 320314, Taiwan; pingyu841215@gmail.com (P.-Y.W.); mei549@gmail.com (M.-F.C.); 4Department of Post-Baccalaureate Medicine, College of Medicine, National Chung Hsing University, Taichung 400, Taiwan; 5Technology Translation Center for Medical Device, Chung Yuan Christian University, Taoyuan 320314, Taiwan; lin_wenchen@cycu.edu.tw (W.-C.L.); clt@cycu.edu.tw (C.-L.T.); 6Department of Biomedical Engineering, Chung Yuan Christian University, Taoyuan 320314, Taiwan

**Keywords:** peptic ulcer, bleeding, computer image, image analysis

## Abstract

The management of peptic ulcer bleeding is clinically challenging. For decades, the Forrest classification has been used for risk stratification for nonvariceal ulcer bleeding. The perception and interpretation of the Forrest classification vary among different endoscopists. The relationship between the bleeder and ulcer images and the different stages of the Forrest classification has not been studied yet. Endoscopic still images of 276 patients with peptic ulcer bleeding for the past 3 years were retrieved and reviewed. The intra-rater agreement and inter-rater agreement were compared. The obtained endoscopic images were manually drawn to delineate the extent of the ulcer and bleeding area. The areas of the region of interest were compared between the different stages of the Forrest classification. A total of 276 images were first classified by two experienced tutor endoscopists. The images were reviewed by six other endoscopists. A good intra-rater correlation was observed (0.92–0.98). A good inter-rater correlation was observed among the different levels of experience (0.639–0.859). The correlation was higher among tutor and junior endoscopists than among experienced endoscopists. Low-risk Forrest IIC and III lesions show distinct patterns compared to high-risk Forrest I, IIA, or IIB lesions. We found good agreement of the Forrest classification among different endoscopists in a single institution. This is the first study to quantitively analyze the obtained and explain the distinct patterns of bleeding ulcers from endoscopy images.

## 1. Introduction

Peptic ulcer bleeding is a common gastrointestinal emergency that requires prompt endoscopic diagnosis and treatment [1,2,3]. Significant progress has been made since the development of various endoscopic therapeutic techniques after the introduction of modern endoscopy [4,5]. Forrest [6] was the first to propose a classification scheme for describing the evolution of such bleeding in 1974. The Forrest classification facilitates communication between different endoscopists and helps in choosing the appropriate endoscopic intervention. However, there is a paucity of data on the performance of interobserver or intra-observer agreement of the old classification. One study conducted in Italy reported high interobserver agreement for Forrest IA/B lesions but low agreement for Forrest II/III lesions [7] utilizing 25 videotapes. As the current endoscopy systems are equipped with high-definition resolution and the images could be stored as digitally captured images, they should be clearer and more suitable for training young endoscopists.

Current guidelines [3,8,9] suggest that patients with high-risk ulcers such as active spurting (Forrest IA), active oozing (Forrest IB), and nonbleeding visible vessel (Forrest IIA) ulcers should undergo endoscopic therapy. Peptic ulcers with adherent clots (Forrest IIB lesion) should be subjected to endoscopic clot removal to decide on further treatment plans. Ulcers with red spots (Forrest IIC) or a clean base (Forrest III) can be observed without endoscopic therapy. However, the Canadian registry of patients with upper gastrointestinal bleeding reported that only 47.8% of patients with high-risk stigmata underwent endoscopic therapy, whereas 9.8% of those with low-risk stigmata underwent endoscopic therapy [10], showing the inconsistency of endoscopist practice in the real world despite the widespread use of the Forrest classification since its introduction.

Thus, in this study, we aimed to compare the intra-rater and inter-rater correlation of the Forrest classification in a modern endoscopy unit. A pixel-by-pixel approach to the obtained endoscopy images was utilized to explore their relationship for the different stages of the Forrest classification.

## 2. Materials and Methods

### 2.1. Patients and Data Preparation

The endoscopy records of patients who underwent endoscopic examinations between January 2017 and January 2020 at the endoscopy center of the Changhua Christian Hospital were retrospectively reviewed. The images were reviewed and retrieved for subsequent analysis by two expert endoscopists (tutors) with 15 years of experience in therapeutic endoscopy. The tutor endoscopists were also involved in training junior endoscopists in the unit. The inclusion criteria for the analysis of the images were (a) images from patients with symptoms of gastrointestinal bleeding, i.e., hematemesis, anemia, or tarry stool; (b) bleeders were attributed to a peptic ulcer disease, i.e., gastric, or duodenal ulcers; and (c) endoscopy performed with an Olympus 260 or 290 series system with digitally stored images.

The study complied with the World Medical Association Declaration of Helsinki for medical research involving human subjects, including research on identifiable human material and data, and was approved by the institutional review board of the Changhua Christian Hospital (approval number: CCH IRB 200906, Approval date: 19 October 2020).

### 2.2. Process of Peptic Ulcer Image Segmentation

The obtained endoscopy images were first cropped in to 476 × 416 pixels per image containing only the endoscopic information of the image. The ulcerated area was defined from the base of the ulcer to its margin (Figure 1). The bleeding area was defined as the culprit bleeder and blood clots adjacent to the bleeder. The area of the ulcer per lesion was defined as the ulcer area/(the ulcer plus bleeder area); the area of bleeder per lesion was defined as the bleeder area/(the ulcer plus bleeder area); and the area of the ulcer and bleeder per lesion was defined as (the ulcer and bleeder area)/(the ulcer plus bleeder area). The region of interest of the ulcer and bleeding area of each obtained image was further drawn with the Fotografix V102 (Available online: https://www.ilsitoblu.com/semplice-alternativa-a-photoshop-fotografix/fotografix102 (accessed on 1 March 2022) and calculated. Two expert endoscopists performed image segmentation.

### 2.3. Evaluation of the Forrest Classification of Peptic Ulcers

The endoscopic images of the bleeding peptic ulcers were first classified according to the Forrest classification, i.e., Forrest I, bleeding or oozing ulcer; Forret IIA, nonbleeding visible vessel, Forrest IIB, bleeding ulcer with adherent blood clots; Forrest IIC, ulcer with pigmentation, and Forrest III, ulcer with a clean ulcer base. The obtained images were reviewed by two expert endoscopists with more than 10 years of experience in therapeutic endoscopy and four junior endoscopists with less than 5 years of experience in the same domain to classify the endoscopy images according to the Forrest classification.

### 2.4. Statistics

Statistical analyses were performed using MedCalc Version 19.8 (2021 MedCalc Software Ltd. 8400 Ostend, Belgium) for interobserver agreement assessments with kappa statistics. The results were classified as follows: poor, ≤0.2; mild, 0.2 to 0.4; moderate, 0.4 to 0.6; good, 0.6 to 0.8; and excellent, 0.8 to 1. A nonparametric test using intraclass correlation was conducted to evaluate the intra-rater agreement. The Kruskal–Wallis test was utilized to compare continuous data with skewed distributions. Results were considered statistically significant if the *p*-value was <0.05.

## 3. Results

### 3.1. Intra-Observer and Interobserver Agreement of the Obtained Images

A total of 112 images, including Forrest I (13.4%), IIA (17%), IIB (6.2%), IIC (21.4%). and III (42%) were utilized for intra-rater agreement for analysis. A total of 276 images, including Forrest I (19.6%), IIA (15.9%), IIB (6.5%), IIC (19.2%), and III (38.8%) were enrolled for interobserver agreement analysis. All the images were stored in a test folder on the desktop and the studying endoscopist separated these images into different Forrest classes during classification.

The intra-rater agreement of the Forrest classification was high for both experienced and junior endoscopists, ranging from 0.92 to 0.97 (Table 1).

The inter-rater agreement of the Forrest classification was good (Table 2), ranging from 0.67 to 0.86. A higher inter-rater agreement was found between tutors and junior endoscopists than between other groups (0.78 vs. 0.68, *p* = 0.01).

### 3.2. Quantitative Analysis Comparing Endoscopic Images from Different Forrest Classifications

Figure 2 illustrates the representative images from Forrest I, IIB, and IIC lesions with the segmented endoscopy images. The ulcerated area was defined from the base of the ulcer to its margin (Figure 1) and the area of the bleeder (i.e., the areas of bleeding vessels and areas of bloody streaks or blood clots) were manually delineated by two experts (Figure 2).

Table 3 illustrates the results of a quantitative analysis of the 276 images. Forrest I, IIA, and IIB have a high bleeder area in the endoscopic image compared with those of low-risk ulcers, i.e., Forrest IIC and III (Figure 3). The clean-based ulcer (Forrest III lesion) contained no bleeder area, whereas the active-bleeding ulcer (Forrest I lesion) contained the highest bleeder area in the obtained endoscopy image. The presence of “adherent blood clots” on Forrest IIB lesions meant a higher bleeder area than that of Forrest IIA lesions with “nonbleeding visible vessels” (68.55% vs. 26.88%, *p* < 0.001). In terms of the overlapping area of the ulcer and bleeder (Figure 4), Forrest I, IIA, and IIB lesions had a similar pattern to that of low-risk ulcers, i.e., Forrest IIC and III.

## 4. Discussion

In this study, we utilized images captured via modern endoscopy to evaluate the performance of the Forrest classification for patients with bleeding peptic ulcers. We found a higher intra-rater agreement of the current system (with an overall mean kappa of 0.71) than was previously described [7]. The inter-rater agreement was higher between tutor endoscopists and junior endoscopists, which could be explained by the teaching effect during the training process [11]. To the best of our knowledge, we are the first to quantitively describe captured endoscopy images of bleeding peptic ulcers and describe their differences. This work paves the way for further development of computer-aided software programs for the endoscopic management of bleeding peptic ulcers in the future.

The Forrest classification was utilized to describe the stigmata of the recent hemorrhage of endoscopic features identified in bleeding peptic ulcers [6]. These endoscopic signs represent different phases of the ulcer in progress. It has a predictive value for the risk of further bleeding, which helps the clinician determine which patients should undergo endoscopic therapy [12]. After moving to the 21st century, there is a significant improvement in medical therapy, such as the widespread use of proton pump inhibitors [13,14,15] and the eradication of *Helicobacter pylori* infections that have led to a decrease in the prevalence of peptic ulcer disease and its complications such as bleeding, perforation, or obstruction [16,17]. A survey conducted in the UK demonstrated that the experience of trainee endoscopists has reduced over the past two decades from 76% in 1996 to 15% in 2011 [2]. Although hemostatic skills such as injection, coagulation, or clipping could be acquired and improved by utilizing ex vivo models [18], experience in determining the optimal management for a bleeding peptic ulcer can typically only be acquired during daily practice in most endoscopy units or offered by some dedicated developed training courses [19,20,21,22]. However, in 1997 Lau [23] reported low agreement between 14 international experts with a weighted kappa of 0.43 on utilizing 100 consecutive edited videotape records from patients with bleeding peptic ulcers. Bour et al. [24] utilized 91 consecutive endoscopic video recordings of adult patients. Nine endoscopists reviewed the videos and the intra-observer agreement was good (kappa = 0.60). These studies were performed in the late 20th century, an era during which the resolution of endoscopy system is low, and the images were only obtained with the use of video tapes or printed images. As far as we know, our study was the first in the 21st century to re-assess the performance of the Forrest classification utilizing a large set of obtained endoscopy images. Both the intra-observer agreement (which ranged from 0.92 to 0.97) and the inter-rater agreement (which ranged from 0.67 to 0.86) of the Forrest classification were higher than previously reported [23,24]. This finding may imply that the superiority of the quality of modern endoscopy images compared to old-fashioned 35-mm slides or videotapes [25] has improved endoscopy training in the 21st century.

One importance of the present study is that we are the first to provide quantitively analyze the different classes of peptic ulcer bleeding. In addition to the patient’s clinical characteristics, the endoscopic appearance of the ulcer provides useful prognostic information on further bleeding [4,12,26]. He et al. [27] recently described the endoscopy morphology change of 298 patients with bleeding gastric ulcer. The study described the size, location and bleeding pattern are significantly correlated with those the stage of the ulcers. Identification of the protruded non-bleeding visible vessel among Forrest IIA ulcer was found to be an independent risk factor of peptic ulcer rebleeding [28]. The endoscopic Forrest classification of artificial gastric ulcers following endoscopic submucosal dissection was useful predicting delayed bleeding [29]. Therefore, careful observation of endoscopic features of all bleeding peptic ulcers helps clinicians in the triage of patients for further treatment plans or, in some cases, in determining the need for surgical interventions.

The difference in the interpretation of endoscopy images highlighted the difference in the perception of each endoscopist’s eyes [13]. To our knowledge, no previous study has attempted to explain the differences in images taken from the different stages of bleeding peptic ulcer. It is impossible to conduct the analysis when the endoscopy images were not electronically captured, and it requires additional manpower to label endoscopy images for analysis. In the present study, we found a distinct pattern of high-risk (Forrest I, IIA, and IIB) and low-risk ulcers (Forrest IIC and III lesions) while analyzing the relationship between the ulcer and bleeder in individual endoscopy images. The ratio of the lowest proportion area of bleeders among Forrest III lesions to the highest proportion area of bleeders among Forrest I lesions corresponds to the evolution of the bleeding ulcer. In addition, Forrest IIB lesions with adherent blood clots appeared to have a higher proportion of bleeder areas than Forrest IIA lesions but like that of Forrest I lesions. This finding provides an explanation of the need to remove adherent blood clots or further clinical decision-making during emergency endoscopy [3,8,9]. Our previous work reported the use of whole endoscopic image to develop an classification deep learning model and found the use of such model have the potential to perform a better prediction than young endoscopist [5]. The present work is the first work of endoscopy image segmentation work for the relationship of bleeding areas and ulcer areas [30,31]. To develop a reliable artificial intelligence system during diagnostic endoscopy, the first work is delineate the lesion for model training [32]. This study of the captured images paves the way for further study on the applications of deep learning methods in this field, i.e., automatic identification possible bleeder, quantification of the ulcer areas or automatic classification of the Forrest classification during endoscopic examination [4,33,34].

However, our study has some limitations. First, all study data came from one hospital in the past five years and only images from the Olympus endoscope system were analyzed. Future studies, including use of the FujiFilm systems will be required to make the study result more generalized and reliable. Second, the classification of the obtained images was done mainly by two tutor endoscopists in our institution, which may not reflect the opinions of other experts from other institutions. Part of patients were assigned to different classes of risk according to the Forrest classification even in expert hands. This may significantly change the prognosis and require further clinical and endoscopically follow-up. Third, unlike endoscopy images taken during screening [31], those from patients with bleeding peptic ulcers are usually taken in an emergency and with less preparation. The obtained images are frequently disturbed by blood clots, food debris, or air bubbles that interfere with the performance of emergency endoscopy. The process of defining an ulcer or bleeder area is sometimes not clear-cut, and we tried to avoid inconsistencies by having two tutor endoscopists review the obtained results in this study.

## 5. Conclusions

We found good agreement in the Forrest classification between endoscopists in a single center. The agreement rate is higher than the ones that were previously reported while utilizing modern high-resolution endoscopy. Our study provides a quantitative analysis of the distinct patterns of bleeding ulcers based on their Forrest classification.

## Figures and Tables

**Figure 1 diagnostics-12-01066-f001:**
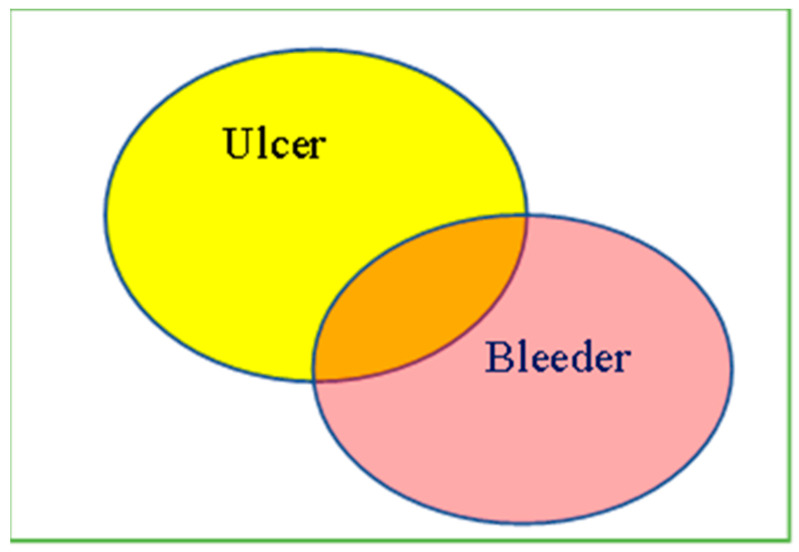
Illustration of Peptic Ulcer Image Processing. The regions of ulcers or bleeders were delineated by two endoscopists. Proportion of Ulcer Area = (Yellow + Orange)/(Yellow + Orang + Pink) × 100%. Proportion of Bleeder Area = (Pink + Orange)/(Yellow + Orang + Pink) × 100%. Proportion of the overlapping area of ulcer/bleeder = (Orange)/(Yellow + Orang + Pink) × 100%.

**Figure 2 diagnostics-12-01066-f002:**
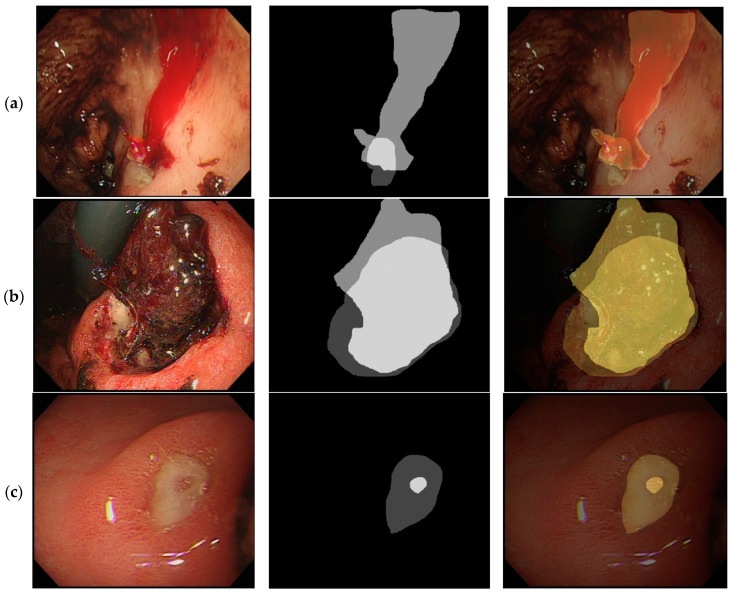
Illustration of the image analysis work of different peptic ulcer images. (**a**). *Forrest*, *I lesion*, actively bleeding ulcer. Original image (**left**), image segmented into ulcer/bleeder areas (**middle**), combined image (**right**). (**b**). *Forrest IIB lesion*, ulcer with adherent clots. Original image (**left**), image segmented into ulcer/bleeder areas (**middle**), combined image (**right**). (**c**). *Forrest IIc lesion*, ulcer with red spots. Original image (**left**), image segmented into ulcer/bleeder areas (**middle**), combined image (**right**).

**Figure 3 diagnostics-12-01066-f003:**
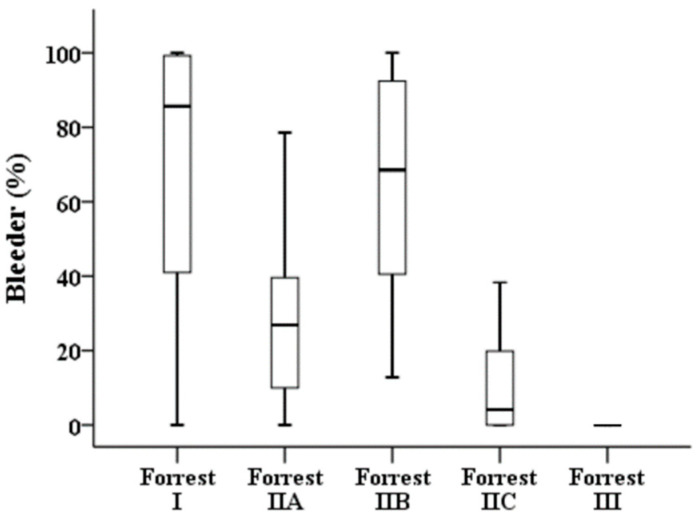
Proportions of bleeding areas in the different stages of the Forrest classification.

**Figure 4 diagnostics-12-01066-f004:**
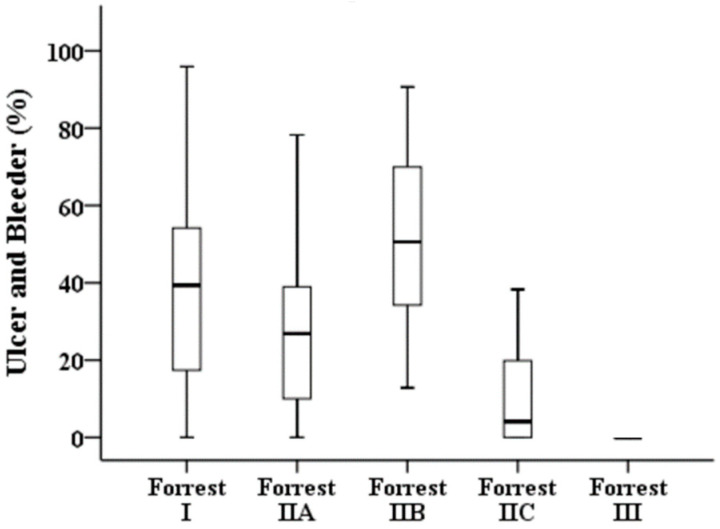
Proportions of overlapping areas of ulcers and bleeders in the different stages of the Forrest classification.

**Table 1 diagnostics-12-01066-t001:** Intra-rater agreement of the Forrest classification.

Rater	Intraclass Correlation	95% Confidence INTERVAL
Experienced 1	0.95	0.94 to 0.97
Experienced 2	0.95	0.93 to 0.97
Junior 1	0.91	0.87 to 0.93
Junior 2	0.92	0.89 to 0.94
Junior 3	0.96	0.95 to 0.98
Junior 4	0.97	0.95 to 0.98

**Table 2 diagnostics-12-01066-t002:** Inter-rater and intra-rater agreements of the Forrest classification.

	Experienced 1	Experienced 2	Junior 1	Junior 2	Junior 3	Junior 4
Tutor ^a^	0.67	0.68	0.74	0.79	0.77	0.79
Experienced 1		0.67	0.66	0.64	0.72	0.71
Experienced 2			0.68	0.64	0.65	0.71
Junior 1				0.67	0.67	0.76
Junior 2					0.86	0.74
Junior 3						0.73

^a^ The consensus result of two expert endoscopists (tutors).

**Table 3 diagnostics-12-01066-t003:** Comparison of bleeder patterns among different Forrest classes of the endoscopy images.

	All Patients (*n* = 276)	Forrest I (*n* = 54)	Forrest IIA (*n* = 44)	Forrest IIB (*n* = 18)	Forrest IIC (*n* = 53)	Forrest III (*n* = 107)
Area of bleeder,%, median (IQR)	4.56 (0–46.33)	85.67 (40.97–99.23)	26.88 (9.99–39.61)	68.55 (40.52–92.4)	4.11 (0–19.87)	0 (0–0)
	Comparison	Forrest I vs. Forrest IIA	Forrest I vs. Forrest IIB	Forrest I vs. Forrest IIC	Forrest I vs. Forrest III	Forrest IIA vs. Forrest IIB
	*p*-value ^a^	0.088	1.000	<0.001	<0.001	<0.001
	Comparison	Forrest IIA vs. Forrest IIC	Forrest IIA vs. Forrest III	Forrest IIB vs. Forrest IIC	Forrest IIB vs. Forrest III	Forrest IIC vs. Forrest III
	*p*-value ^a^	0.019	<0.001	<0.001	<0.001	<0.001
Overlapping area of ulcer and bleeder,%, median (IQR)	4.22 (0–35.07)	39.38 (17.4–54.19)	26.88 (9.99–38.94)	50.57 (34.22–69.94)	4.11 (0–19.87)	0 (0–0)
	Comparison	Forrest I vs. Forrest IIA	Forrest I vs. Forrest IIB	Forrest I vs. Forrest IIC	Forrest I vs. Forrest III	Forrest IIA vs. Forrest IIB
	*p*-value ^a^	1.000	1.000	<0.001	<0.001	0.536
	Comparison	Forrest IIA vs. Forrest IIC	Forrest IIA vs. Forrest III	Forrest IIB vs. Forrest IIC	Forrest IIB vs. Forrest III	Forrest IIC vs. Forrest III
	*p*-value ^a^	0.007	<0.001	<0.001	<0.001	<0.001

^a^ With Dunn-Bonferroni post hoc method to compare the *p* value between five different Forrest classification. The *p* value was <0.001 of the five Forrest classification tested by Kruskal-Wallis Test.

## Data Availability

The data is available on reasonable request.

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
