# Peer review of "Forrest Classification for Bleeding Peptic Ulcer: A New Look at the Old Endoscopic Classification"

_diagnostics, 2022, doi:10.3390/diagnostics12051066_

Round 1

Reviewer 1 Report

This is an interesting study aiming at defining the intra and inter observer variation in the assessment of the Forrest Classification of Ulcer Bleeding. They have shown an high intra and a good inter observer agreement. Moreover, they tried to introduce an additional criterion, that is the proportion of the bleeding are in the context of the ulcer.

This is an interesting paper, that needs refinement:

  1. Table 2 needs clarification: what is meant with Tutor, the sum of the two experienced endoscopist? please clarify.
  2. A defined cut off of of inter observer agreement should be introduced: as I understand it, one third of patients were assigned to different classes of risk according to the Forrest classification in expert hands, and this may change significantly the prognosis and hence the further clinical and endoscopical follow-up. Please discuss as limitation 
  3. 2. As I read it, there was a greater agreement among less experienced endoscopists than between the two experienced ones: if this is true, please explain, if not, make it clearer
  4. fixed, single images may not be better that videos to assess a bleeding ulcer; videos, in fact may better describe, dynamically, the area of interest. This should be discussed.
  5. definition of the area of interest is intriguing, but I do not see any clinical consequence coming ut of this. Could you reassess outcome in relation to this? It would be very interesting
  6. The potential role of image capturing could be better put in perspective of the application of artificial Intelligence in this context: this can be discussed and made clearer

Author Response

Dear Reviewer,

Thank you for reviewing our manuscript and providing your editorial comments as well as reviewers comments for improving our manuscript. Based on these comments, we have made several revisions to our manuscript, which we are hereby resubmitting for your consideration. Our point-by-point responses to the comments are detailed below.

Reviewer 1

#1. This is an interesting study aiming at defining the intra and inter observer variation in the assessment of the Forrest Classification of Ulcer Bleeding. They have shown an high intra and a good inter observer agreement. Moreover, they tried to introduce an additional criterion, that is the proportion of the bleeding are in the context of the ulcer. This is an interesting paper, that needs refinement:

Table 2 needs clarification: what is meant with Tutor, the sum of the two experienced endoscopist? please clarify.

Response: Thank for your comment. The result of Tutor referred to the consensus result of two expert endoscopists (tutors). We add explanation in the revised manuscript Table 2.,

#2. A defined cut off of of inter observer agreement should be introduced: as I understand it, one third of patients were assigned to different classes of risk according to the Forrest classification in expert hands, and this may change significantly the prognosis and hence the further clinical and endoscopical follow-up. Please discuss as limitation

Response: Thank for your comment. We agreed there is no perfect classification even in expert hands, we add this as a limitation of the study.

#3. As I read it, there was a greater agreement among less experienced endoscopists than between the two experienced ones: if this is true, please explain, if not, make it clearer fixed, single images may not be better that video to assess a bleeding ulcer; videos, in fact may better describe, dynamically, the area of interest. This should be discussed. definition of the area of interest is intriguing, but I do not see any clinical consequence coming ut of this. Could you reassess outcome in relation to this? It would be very interesting

Response: Thank for your comment. The result is true and interesting. As we mentioned in the discussion, we believe the result could be explained by the teaching effect during the training process. This is also why the clinicians require continued medical education even they are experienced.

#4. The potential role of image capturing could be better put in perspective of the application of artificial Intelligence in this context: this can be discussed and made clearer

Response: Thank for your comment. We add discussion in the revised manuscript.

Thank you for the opportunity to resubmit this manuscript for consideration for publication in the Diagnostics If you have any questions or comments regarding this manuscript, please do not hesitate to contact us using the details provided below.

Sincerely,

Hsu-Heng Yen M.D

Division of Gastroenterology, Department of Internal Medicine, Changhua Christian Hospital, Changhua, Taiwan

Fax: +886-4-7228289

Tel: +886-4-7238595ext5501

E-mail: 91646@cch.org.tw, blaneyen@gmail.com

Reviewer 2 Report

(1) Table 3 is a little difficult to understand and took a little extra effort to understand. I would recommend making it a little easier to understand.
(2) Bring consistency in intrarater and interrater words. I suggest a "-" every time you use these words which will make it easy for readers, like intra-rater and inter-rater.
(3) Self-citations did not impress me. I recommend avoiding it.
(4) Citations have been overused. I suggest cutting down on unnecessary citations.
(5) A higher number of providers is suggested for any future studies to make it more generalized and reliable. Also, a lot of places use FujiFilm scopes, and including them in future studies will definitely be more appealing.

Author Response

Dear Reviewer,

Thank you for reviewing our manuscript and providing your editorial comments as well as reviewers comments for improving our manuscript. Based on these comments, we have made several revisions to our manuscript, which we are hereby resubmitting for your consideration. Our point-by-point responses to the comments are detailed below.

Response to Reviewers’ comments

Reviewer #2.

  • Table 3 is a little difficult to understand and took a little extra effort to understand. I would recommend making it a little easier to understand. Bring consistency in intrarater and interrater words. I suggest a "-" every time you use these words which will make it easy for readers, like intra-rater and inter-rater.

Response: Thank for your comment. We improve the readability of the revised manuscript according to your suggestion.

  • Self-citations did not impress me. I recommend avoiding it. Citations have been overused. I suggest cutting down on unnecessary citations.

Response: Thank for your comment. We cut down on unnecessary citations in the revised manuscript.

  • A higher number of providers is suggested for any future studies to make it more generalized and reliable. Also, a lot of places use FujiFilm scopes, and including them in future studies will definitely be more appealing.

Response: Thank for your comment. We add your comment in the revised manuscript.

Thank you for the opportunity to resubmit this manuscript for consideration for publication in the Diagnostics If you have any questions or comments regarding this manuscript, please do not hesitate to contact us using the details provided below.

Sincerely,

Hsu-Heng Yen M.D

Division of Gastroenterology, Department of Internal Medicine, Changhua Christian Hospital, Changhua, Taiwan

Fax: +886-4-7228289

Tel: +886-4-7238595ext5501

E-mail: 91646@cch.org.tw, blaneyen@gmail.com

Reviewer 3 Report

The paper "Forrest Classification for Bleeding Peptic Ulcer: A New Look at 2 the Old Endoscopic Classification" is an excellent well-written and documented experience on the evaluation of Forrest classification with new methods of endoscopic investigation. The work is excellent with excellent bibliographic and iconographic documentation worthy of publication.

Author Response

Dear Reviewer,

Thank you for reviewing our manuscript and providing your editorial comments as well as reviewers comments for improving our manuscript. Based on these comments, we have made several revisions to our manuscript, which we are hereby resubmitting for your consideration. Our point-by-point responses to the comments are detailed below.

Reviewer #3:

 The paper "Forrest Classification for Bleeding Peptic Ulcer: A New Look at 2 the Old Endoscopic Classification" is an excellent well-written and documented experience on the evaluation of Forrest classification with new methods of endoscopic investigation. The work is excellent with excellent bibliographic and iconographic documentation worthy of publication.

Response:

Thank for your comment.

Thank you for the opportunity to resubmit this manuscript for consideration for publication in the Diagnostics If you have any questions or comments regarding this manuscript, please do not hesitate to contact us using the details provided below.

Sincerely,

Hsu-Heng Yen M.D

Division of Gastroenterology, Department of Internal Medicine, Changhua Christian Hospital, Changhua, Taiwan

Fax: +886-4-7228289

Tel: +886-4-7238595ext5501

E-mail: 91646@cch.org.tw, blaneyen@gmail.com